# A Thermal Infrared Pedestrian-Detection Method for Edge Computing Devices

**DOI:** 10.3390/s22176710

**Published:** 2022-09-05

**Authors:** Shuai You, Yimu Ji, Shangdong Liu, Chaojun Mei, Xiaoliang Yao, Yujian Feng

**Affiliations:** 1School of Internet of Things, Nanjing University of Posts and Telecommunications, Nanjing 210023, China; 2Institute of High Performance Computing and Bigdata, Nanjing University of Posts and Telecommunications, Nanjing 210023, China; 3School of Computer Science, Nanjing University of Posts and Telecommunications, Nanjing 210023, China; 4Nanjing Center of HPC China, Nanjing 210023, China; 5Jiangsu HPC and Intelligent Processing Engineer Research Center, Nanjing 210023, China

**Keywords:** pedestrian detection, thermal infrared images, attention mechanism, data augmentation, real-time

## Abstract

The thermal imaging pedestrian-detection system has excellent performance in different lighting scenarios, but there are problems regarding weak texture, object occlusion, and small objects. Meanwhile, large high-performance models have higher latency on edge devices with limited computing power. To solve the above problems, in this paper, we propose a real-time thermal imaging pedestrian-detection method for edge computing devices. Firstly, we utilize multi-scale mosaic data augmentation to enhance the diversity and texture of objects, which alleviates the impact of complex environments. Then, the parameter-free attention mechanism is introduced into the network to enhance features, which barely increases the computing cost of the network. Finally, we accelerate multi-channel video detection through quantization and multi-threading techniques on edge computing devices. Additionally, we create a high-quality thermal infrared dataset to facilitate the research. The comparative experiments on the self-built dataset, YDTIP, and three public datasets, with other methods show that our method also has certain advantages.

## 1. Introduction

Pedestrian object detection is an important research direction in computer vision, and consists of feature extraction, object classification, and object localization. Most of the existing methods are mainly concerned with object detection under visible light. However, due to complex lighting conditions, e.g., cloud, rain, fog, etc., visible light methods are insufficient for extracting features. Hence, infrared cameras can be used, which can sense the radiation of the object and can effectively avoid the interference of complex scenes during detection [1]. However, thermal infrared images have the disadvantages of low contrast, weak texture, and loud noise. These physical properties still lead to challenging object detection in thermal infrared scenes [2,3].

Infrared pedestrian-detection methods can be divided into traditional methods and convolutional neural network (ConvNet) methods [4]. Feature extraction and classification are performed separately in traditional object detection. Firstly, the pedestrian features (e.g., color, edge, texture, etc.) are extracted by handcraft; then, the classifier is trained with these features; finally, sliding windows are utilized for object localization [5]. Compared with traditional detection methods, ConvNets can implement end-to-end pedestrian detection, which is independent of expert experience [6]. The ConvNet methods consist of a feature extraction network, bounding-box regression, and object classification [7].

The traditional methods are based on handcraft features [8]. Zhang et al. [9] designed rectangular templates of the human body with multi-modal and multi-channel Haar-like features. Watanabe et al. [10] utilized two co-occurrence histogram features (i.e., CoHOG and CoHLBP), which describe different aspects of object shapes to improve the accuracy of pedestrian detection. Felzenszwalb et al. [11] proposed latent SVM, which combines a margin-sensitive method for data mining hard negative examples. Zhang et al. [12] extended two feature classes (edgelets and HOG features) and two classification models (AdaBoost and SVM cascade) to infrared images. Lowe et al. [13] introduced a scale-invariant feature transform for object recognition. Similarly, Dalal et al. [14] utilized the histogram of an oriented gradient for feature extraction. Professor Maldague’s team utilized visible light and thermal infrared image pairs to extract regions of interest [15]. However, these methods have insufficient performance in complex scenarios. Moreover, the strategy of sliding windows generates a massive number of redundant boxes. Additionally, the strategy of handcrafted features lacks high-level semantic information. These two strategies lead to low object detection efficiency [16].

The ConvNet methods involve the Yolo series [17] and the SSD series [18], which have huge parameters (i.e., the weight parameters of the network) and high latency on edge computing devices. The edge computing devices for these series are only Internet-of-Things terminal devices, which have low computing power and power consumption [19]. Hence, the MobileNet [20] series and the ShuffleNet [21] series were presented, which are represented by a lightweight base model [22]. Zheng et al. [23] utilized MobileNetv3 and depthwise separable convolution to achieve parameter compression and inference acceleration of Yolov4 (i.e., Yolov4-lite). Wang et al. [24] achieved SSD parameter compression using MoblieNetV2 in infrared image pedestrian detection. There is also Yolov4-tiny [25], which performs lightweight reconstruction of Yolov4. According to the scenario’s needs, EfficientDet [26], and Yolov5 [27] utilized hyperparameters to control the network parameters. For example, the network structure of EfficentDet (i.e., D0 to D7) is, respectively, associated with the input size, where D0 corresponds to the input size of 512 × 512. Rep-Yolov4 [28] utilized network pruning to compress network parameters, speed up inference, etc.

The above ConvNet methods (i.e., anchor-based object detectors) are less effective for some specific datasets, which require the reclustering of anchors [29]. Many studies have shown that anchor-free detectors can perform comparably to anchor-based detectors in recent years. The Yolox of anchor-free detectors introduces a decoupled head, which reduces the conflict between classification and regression tasks [30]. Liu et al. [4] proposed a thermal infrared pedestrian and vehicle detector based on Yolov3 by expanding the parameters of YoloHead, which consist of anchor-free and anchor-based. To enhance the features, the attention mechanism was introduced to improve detection performance, e.g., SENet [31] and CBAM [32]. Xue et al. [33] proposed a spatio-temporal sharpening attention mechanism, which combined the channel attention module and the sharpening spatial attention module. Xue et al. [34] proposed novel multi-modal attention fusion based on Yolo for real-time pedestrian detection.

However, due to the small coverage area, low resolution, and reduced feature information of small objects in the thermal infrared images, the performance of the above methods is unsatisfactory. Gao et al. [35] proposed a high-precision detection method based on the feature mapping of deep neural networks with a spindle network structure. Lu et al. [36] proposed a novel small-object detection method via multidirectional derivative-based weighted contrast measures. Zhu et al. [37] introduced an attention mechanism based on RefineDet for the infrared detection of small pedestrians. Li et al. [38] utilized the weighted fusion of visible images and infrared images, which enhanced small-feature information. In this paper, we proposed an efficient thermal infrared detection method based on Yolox-S in complex pedestrian scenes. Our method can effectively solve the problem of small objects and occlusion objects, without adding any parameters. To sum up, our main contributions are:(1)We adjust the input size of the network, which reduces the calculation redundancy and alleviates the imbalanced number of small objects based on the self-built dataset.(2)Multi-scale mosaic data augmentation is proposed to extend object diversity in training, which simulates multiple objects, small objects, and occlusion objects scenarios.(3)To achieve effective feature fusion in the neck, we introduce the parameter-free attention mechanism to implement feature enhancement.(4)We accelerate the network inference using quantization technology on the edge device. To fully exploit hardware resources, multi-threading technology is utilized to realize multi-channel video detection tasks.

The rest of this paper is organized as follows. Section 2 details the proposed method. Section 3 conducts the experiments based on the infrared image datasets (YDTIP, UTMD [39], KAIST [40] and CVC-14 [41]) to verify the effectiveness of the proposed method. Section 4 summarizes this paper.

## 2. Proposed Method

In recent years, the ConvNet structure has been iteratively updated, and the capability of feature extraction has become stronger. In this paper, the main improvements based on Yolox-S in thermal imaging pedestrian-detection (TIPYOLO) are as follows: we modify the size of the input image by analyzing the characteristics of the dataset, and increase the diversity of the training set through a multi-scale mosaic; the parameter-free attention mechanism is introduced into Yolox-S; we utilize quantization and multi-threading technology to meet business timeliness requirements.

### 2.1. Multi-Scale Mosaic

In the object detection network, the input of the network has a uniform size, and generally has a square structure. The customary methods obtain input sizes of the standard by cutting, stretching, scaling, etc. However, these operations are prone to problems such as missing objects, a loss of resolution, and reduced accuracy. In the testing stage of Yolox-S, the sizes of the input images are 416 × 416 or 640 × 640. Hence, we utilize gray filling to achieve image scaling in the model inference phase.

Here, we take an image of 416 × 416 as an example; in Figure 1, A is the original input image. After calculating the scaling ratio of the A image, image B is obtained using the BICUBIC interpolation algorithm. Image B is pasted into a 416 × 416 gray image to obtain C1. This method of image scaling reduces the image resolution without distortion, but the filled gray pixel area in C1 is too large (i.e., more than half of the gray-filled area, causing redundant computation), which increases the inference time of the model. Meanwhile, considering the limited resources of edge computing devices, the maximum downsampling rate of the ConvNet is 32 times, which results in an optimal gray filling scale of 416 × 192, as shown by C2 in Figure 1.

For the model training phase, we introduce mosaic data augmentation [17]. Mosaic data augmentation mainly crops four pictures in a central position, and then, merges them into one picture after cropping. As shown in Figure 2a,b, the mosaic can enrich the scene, which involves the image background, small objects, and pedestrian occlusion. Moreover, a larger batchsize will increase the GPU memory burden. The Mosaic operation can reduce the dimension of the input data, that is, channel × height × width (CHW) × batchsize is reduced to CHW × batchsize × 1/4, thereby reducing the memory usage. This is due to the mosaic data augmentation being away from the distribution of natural images. Therefore, the code automatically turns off the mosaic during the training of the last 30 epochs, and the random probability of training with mosaic is 0.5.

On the basis of the above strategies, we utilize a random multi-scale training strategy. Since the maximum downsampling rate of the network is 32 times, the size sequence of the input image is also selected with an integer multiple of 32, that is, 288 × 64, 320 × 96, 352 × 128, 384 × 160, 416 × 192, 480 × 224, 512 × 256, 544 × 288, 576 × 320, 608 × 352, and 640 × 384. A scale is randomly selected in the sequence as the input size. This strategy can not only improve the adaptability of the network to different scales, but can also realize the scale perturbation of objects.

### 2.2. TIPYOLO Network Details

TIPYOLO is an improvement on Yolox-S in the thermal imaging of pedestrian detection. The input of model is a three-channel RGB image (i.e., 416 × 192 × 3). The backbone consists of CSPDarknet, which integrates general residuals and cross-stage partial residuals. Hence, CSPDarknet achieves efficient feature extraction with few parameters. As shown in Figure 3, SPPBottleneck is added to CSPDarknet, which mainly plays the role of multi-receptive field fusion. Between the backbone and neck, we introduce the parameter-free attention module (i.e., simAM) [42]. We utilize simAM to filter more focused feature information for the neck.

The theory of simAM is based on visual neuroscience. In visual processing, neurons with clear spatial suppression effects should be given higher priority. To better implement the attention mechanism, the simAM algorithm evaluates the importance of each neuron using the minimum energy equation el*. The lower the energy, the more distinctive the target neuron from the surrounding neurons, and the more important it is for visual processing. The importance of each neuron can be obtained using 1/el*, as shown in Equation (1).
(1)1/el*=(t−u^)2/4(σ^2+λ)+0.5
where u^=1/M∑i=1Mxi and σ^2=1/M∑i=1M(xi−u^)2; t and xi are the target neuron and other neurons in a single channel of the input feature X∈ℝC×H×W; i is index over spatial dimension; and M=H×W is the number of neurons on that channel. The hyper-parameter λ is set to 0.0001

According to the definition of an attention mechanism, we need to enhance the feature, as shown in Equation (2).
(2)$$\tilde{X}=sigmoid(1/E)⊙X$$
where E groups all of $e_{l}{}^{*}$ across channel and spatial dimensions.

PANet performs the fusion features of bottom-up and top-down; these features are filtered by the attention module. Here, feature fusion adopts Concat in the channel dimension, which has rich semantic information, that is, it includes both network deep features and shallow features. The fused features are passed to YoloHead for object classification and location regression. The first Conv layer of YoloHead mainly implements channel adjustment, which reduces the channel size and the amount of branch computation. The two parallel branches (i.e., localization and classification) have two 3 × 3 Conv layers, respectively, which further improves the nonlinear fitting ability of the network.

### 2.3. Video Processing Acceleration

During the deployment phase of the method, we utilize post-training quantization techniques to achieve network inference acceleration on edge computing devices. The model quantization is the process of mapping float values to int values. We take int8 as an example, to map the floating-point input to the fixed point. First, we need to count the numerical range of the input, which provides the maximum value of its absolute value (i.e., |max|). Then, |max| is mapped to 127 (i.e., the maximum value under int8), obtaining the mapped ratio $s^{1}$ using Equation (3).
(3)$$s^{1}=\frac{127}{\left|\max\right|}$$

$s^{1}$ is the scaling ratio, which is the floating-point input mapped to the fixed point. Similarly, the weight scaling ratio $s^{2}$ can be calculated. Finally, $s_{i}^{1}$ and $s_{i}^{2}$ are saved in the weight parameter for subsequent fixed-point calculations, where the index i is the current network layer.

Here, we take the convolution operator as an example, as shown in Figure 4. First, the floating-point input is quantized to obtain an integer input using the scaling ratio (i.e., $s_{i}^{1}$ and $s_{i}^{2}$), as shown in Equations (4)–(6).
(4)$$x^{\prime }=round(x*s_{i}^{1})$$
(5)$$clamp(x^{\prime },|T|)=\left\{ \begin{array}{l} x^{\prime },\;\;\;\;\;|x^{\prime }|\leq |T|\\ sign(x^{\prime })\times |T|,\;|x^{\prime }|>|T| \end{array}\right.$$
(6)$$qx=clamp(x^{\prime },|T|)$$
where x is the input, and qx is the quantized result. T is the threshold, and its value range is [−127, 127]. Similarly, the weight of ConvNet is quantized to obtain an integer weight. Then, an integer Conv operation is performed to obtain an output for the integer result. Finally, the floating-point output is obtained through the dequantization Equation (7), where qy is the result of the Conv operation, and y is the dequantized result [43].
(7)$$y=\frac{qy}{s_{i}^{1}\times s_{i}^{2}}$$

The above quantification algorithm is mainly utilized for network inference acceleration, which neglects the calculation acceleration of pre-processing and post-processing. The pre-processing, inference, and post-processing of the model are serial executions. Inevitably, some units of computation will be waiting. To alleviate the wait of series execution, we convert the three computational units of the model into pipeline executions, as shown in Figure 5b, that is, each computing unit is assigned a thread. As shown in Figure 5c, unlike b, two threads are added to speed up the pre-processing. However, pipelineV2 only works in single-channel video processing. For multi-channel video processing, we take two-channel video as an example, and the specific strategy is shown in Figure 5d,e. As shown in Figure 5d, we utilize pipelineV2 to implement parallelism processing of the two-channel video. However, considering that the two models are running on edge computing devices at the same time, which inevitably increases the hardware burden, we implement two-channel video processing based on the improved pipelineV2. The specific implementation details are: First, we utilize two buffer queues to temporarily store the pre-processing data. Then, the model reads the data alternately from the two queues. Once model inference is completed, the video data of post-processing are stored alternately in two buffer queues using the Flag operation. Finally, the visualization thread utilizes two queue datapoints to visualize.

## 3. Experiments and Results

### 3.1. Data Set and Experiment Setting

Currently, most of the public thermal infrared pedestrian datasets have relatively simple meteorological environments and low resolution. Hence, we generated a high-resolution thermal infrared pedestrian dataset (i.e., YDTIP), which was collected at a school. Additionally, YDTIP contains a rich meteorological environment, which involves cloud, rain, day-time, and night-time. The resolution of each image is 2624 × 1080 in the dataset. The dataset was collected using the automatic thermal imaging temperature measurement system TIC400B, the view field of system was about 25° × 19°, and the lens focal length was 18 mm.

During the COVID-19 epidemic, the system has been widely utilized for temperature measurement in crowded places. The thermal infrared images of pedestrians in different environmental scenarios are shown in Figure 6. As can be seen from the figure, infrared imaging is insensitive to interference by environmental factors.

First, we extracted interval frames from the video, then we cleaned them, and finally, we obtained 3193 pictures. We utilized LabelImg to manually label these images, and only labeled the objects whose occlusion was less than 80%, that is, heavy occlusion (35~80%), partial occlusion (1~35%), no occlusion (0%) [44]. The label data were saved in VOC format (i.e., XML). According to coco API’s definition of large, medium, and small objects, a small object is defined as an object with pixels less than 32 × 32, a medium object is defined as an object with pixels greater than 32 × 32 and less than 96 × 96, and a large object is defined as an object with pixels greater than 96 × 96 [45]. Here, the definition is an absolute concept. Hence, we scaled the long edge of the images to 416; the generated data distribution of large, medium, and small objects is shown in Figure 7a, of which there are 7847 small objects, 7133 medium objects, and 977 large objects.

At the University of Tokyo, Japan, researchers generated a multispectral dataset for autonomous vehicles. Multispectral images are composed of RGB images, near-infrared images, middle-infrared images, and far-infrared images, corresponding to 7512 images, respectively. Since the multispectral dataset contained five classes (i.e., bike, car, car_stop, color_cone, and person), the experiments only utilized the person class. According to the person labels, 2537 images were obtained from middle-infrared images. The data distribution is shown in Figure 7b, of which there are 2719 small objects, 1149 medium objects, and 232 large objects, namely UTMD. It should be noted that each image has a size of 320 × 256 in the UTMD dataset [39]. For the size characteristics of this dataset, during training, since the maximum downsampling rate of the network was 32, the input image size sequence was also selected with an integer multiple of 32, that is, 224 × 160, 256 × 192, 288 × 224, 320 × 256, 352 × 288, and 384 × 320.

We utilized the Pytorch deep learning framework for experiments under the Ubuntu 18.04 system. The machine configuration utilized in this experiment was as follows: the processor was AMD Ryzen 5 3500X 6-Core; the memory was 16GB; and the graphics card was Nvidia 1080Ti (11GB). The software environment utilized was as follows: Pytorch 1.8.1 + cu102, Python 3.6.13.

The model training adopts the transfer learning method. The model loads pre-trained weights on the VOC dataset, which can speed up the convergence of model training. Hence, this strategy is widely adopted. The thermal imaging dataset has a drab object color. The model is prone to overfitting due to only using the original image during the training phase. Hence, we added image horizontal flipping (IHF) and length and width scaling (LWS), as shown in Figure 8b. Then, we added color gamut transformation on the HSV color space (CGT-HSV), as shown in Figure 8c. Moreover, according to the distribution of the RGB three-channel pixel histogram in Figure 8d–f, it can be seen that the pixel difference between the three images is obvious. This is especially true for the augmentation operation in Figure 8f, which can further highlight the object contour information. This data augmentation strategy was utilized to increase the diversity of the dataset and reduce the risk of model overfitting. When the model was tested, only the gray pixels were filled. The YDTIP dataset is divided into 1564 images for training, 671 images for validation, and 958 images for testing. The UTMD dataset is divided into 1242 images for training, 533 images for validation, and 762 images for testing. Meanwhile, we also trained and tested on the commonly used KAIST [40] and CVC-14 [41] datasets, corresponding to 3050 and 2936 images, which performed similar split operations of the dataset. The image size of the KAIST dataset is 640 × 512, and the image size of the CVC-14 dataset is 640 × 480.

During training, we utilized BCE loss for the cls and obj branch, and IoU loss for the reg branch [24]. Model training was completed in two stages. In the first stage, we froze the backbone network for training. In this phase, we set the batchsize to 16, the learning rate to 0.001, the momentum to 0.937, and the weight decay to 0.0005, and a total of 50 epochs were trained. In the second stage, we unfroze the backbone network for training, set the batchsize to 8, and also trained for 50 epochs. SGD (stochastic gradient descent) was utilized as the optimizer of the whole training process. The learning rate was adjusted utilizing a cosine schedule. In comparison experiments, these training hyper-parameters were the same as the settings of each detector.

### 3.2. Evaluation Indexes

To accurately evaluate the performance of the proposed method, we selected four generally utilized evaluation indexes: frames per second (FPS), mean average precision (mAP), log-average miss rate, and parameters. FPS was utilized as the evaluation index of the model inference speed, the mAP was utilized as the evaluation index of the model accuracy, and the parameters were utilized as the evaluation index of the model memory occupation. Precision represents the proportion of correctly detected persons in all positive detection results, and Recall represents the proportion of correctly detected persons in the ground-truth box. The definitions are expressed as Equations (8) and (9), respectively.
(8)$$\mathrm{Precision}=\frac{TP}{TP+FP}$$
(9)$$\mathrm{Recall}=\frac{TP}{TP+FN}$$

TP (true positive) represents the number of correct predictions for all positive samples, FP (false positive) represents the number of incorrect predictions for all negative samples, and FN (false negative) represents the number of incorrect predictions for all positive samples. The average precision (AP) was calculated by utilizing the integral area of the Precision–Recall curve; Recall and Precision are abscissa and ordinate, respectively [46]. The area enclosed by the curve is the value of AP, as in Equation (10). The larger the area, the higher the detection accuracy. AP_50_ is the average precision calculated when the IoU is 0.5, and AP_75_ is similar. AP is the average of the precision, calculated by IoU at intervals of 0.05 from 0.5 to 0.95. AP_S_ represents the evaluation accuracy of small objects by IoU at intervals of 0.05 from 0.5 to 0.95. AP_M_ and AP_L_ are similar [47].
(10)$$\mathrm{AP}=\int_{0}^{1}P(R)dR$$

The log-average miss rate is the evaluation index of detectors corresponding to AP [48]. The miss rate (MR) can be calculated using the number of TPs and the number of ground-truths, as in Equation (11). Additionally, the false positives per image (FPPI) can be calculated by dividing FP by the number of images, as in Equation (12). The log-average MR is calculated by averaging the MRs under 9 FPPI equally spaced in the range of [0.01: 1] (i.e., MR^−2^). Conceptually, the log-average MR is similar to AP for object detection, in that they both use a reference value to represent the entire curve, that is, the Precision-Recall curve and the MR–FPPI curve. When evaluating detectors using MR^−2^, a lower MR^−2^ indicates better performance [44].
(11)$$\mathrm{MR}=\frac{FN}{TP+FN}=1-\mathrm{Recall}$$
(12)$$\mathrm{FPPI}=\frac{FP}{N}$$
where N is the number of images.

### 3.3. Analysis of Experimental Results

#### 3.3.1. Ablation Study on the YDTIP Dataset

Here, we explain the reason for choosing the 416 × 416 input size. According to the definition of COCO API, we scaled the image’s long side to 640. At this size, the number of small objects was 3221, the number of medium objects was 9907, and the number of large objects was 2829, leading to the number imbalance problem. Small-object detection is a difficult task in the object detection field. Hence, we scaled the image’s long side to 416, the number of small objects was 7847, the number of medium objects was 7133, and the number of large objects was 977. Compared with 640, the imbalance problem of small and medium objects under the ratio of 416 was alleviated. Moreover, it can be seen from Table 1 that the detection of large objects at different scales has good performance. Meanwhile, from Table 1, we can see that the quantitative analysis of detector performance utilizing AP, AP_50_, and AP_75_ yields good results under 640 × 640 conditions. However, the performance is poor under the AP_S_ index, which may be caused by insufficient training data for small objects. Additionally, it can be seen from Table 1 that the overall performance of the detector improves slightly compared to 416 × 416 when the input is 416 × 192, which further shows that less gray redundant calculation is necessary.

To prove the effectiveness of each improvement strategy, we conducted ablation experiments on the YDTIP dataset, and the results are shown in Table 2. The operators were: spatial (GAP) or channel average pooling (CAP), spatial (GMP) or channel max pooling (CMP), standard convolution (C2D) or channel-based convolution (C1D), standard fully connected layers (FC), batch normalization (BN), and ReLU. k and r are convolutional filter numbers and the reduction ratio, respectively. C is the current feature channels.

We compared the simAM module with three representative attention modules, SE [31], CBAM [32], and ECA [49], respectively. The specific evaluation results are shown in Table 2 (rows 2, 3, 4, and 5). From Table 2, we can see that the performance of each attention is slightly degraded under some indexes, but the overall increase is still relatively large. It is worth noting that the simAM module does not add any parameters to the existing network, which has a great advantage over other modules.

To more intuitively view the impact of each attention module, we utilized Grad-CAM [50] to visualize the focus heatmap of each detector. As shown in Figure 9, the brighter the color of the heatmap, the more the detector pays attention to the feature. By observation, it is found that the attention area of simAM is larger than other attention modules on the object.

For the data augmentation ablation experiment, it can be seen in Table 2 (rows 1, 6, and 7) that the multi-scale mosaic has a significant increase compared with the baseline (i.e., Yolox-S). Additionally, the multi-scale mosaic, compared with the mosaic, AP, and AP_75_, increased by 0.1% and 0.2%, respectively, and AP_M_ and AP_L_ increased by 0.4% and 1.7%. This further proves that the multi-scale mosaic strategy can effectively improve dataset diversity and model performance.

The simAM and multi-scale mosaic strategies were effectively combined, and the performance evaluation results are shown in Table 2 (last row). Compared with the baseline, the combined performance is significantly improved; AP and AP_75_ are improved by 0.6% and 1%, and AP_S_ is improved by 0.8%. The activation heatmap of TIPYOLO is shown in Figure 8. From the figure, we can see that our method has a larger receptive field for the object and a higher focus on the global features.

In summary, the results of the ablation experiments show that our strategies significantly improve the performance of the detector without introducing any parameters.

#### 3.3.2. Comparison Experiments with Other Methods on the YDTIP Dataset

To further verify the performance of the improved method, TIPYOLO was compared with other methods on the YDTIP dataset, and the comparison results are shown in Table 3.

The experimental results show that TIPYOLO is better than Yolov5-S in terms of accuracy, but the speed is slightly slower by about 5 FPS. The FPS of TIPYOLO is nearly two times worse than the lightweight method Yolov4-tiny, but in terms of accuracy, TIPYOLO is better overall than Yolov4-tiny. Although the TIPYOLO parameter is nearly 2.3 times that of the lightweight method EfficientDet-d0, both AP and FPS are better than EfficientDet-d0. The reason EfficientDet-d0 has few parameters and low speed may be due to more feature fusion and more network branches, and the current deep learning framework has not further accelerated its inference. Compared with Yolov4, the performance of TIPYOLO is 0.1% less than Yolov4 on AP_50_ and AP_S_. Compared with Yolov3, TIPYOLO is 0.2% less than Yolov3 on AP_50_. But in FPS and parameters indexes, our method is better than Yolo3 and Yolov4. Meanwhile, the performance of our method is overall better than SSD.Yolov4-lite utilizes depthwise separable convolution to replace the standard 3 × 3 convolution, and realizes the parameter compression and inference speed improvement of Yolov4, but the accuracy also decreases. Compared with Yolov4-lite, TIPYOLO achieves the best performance overall. In terms of AP indicators, TIPYOLO is lower than Faster-RCNN, but it still has advantages in terms of FPS and parameters. On the MR^−2^ indicator, TIPYOLO and the three large models (i.e., Faster R-CNN, YOLOv3, and YOLOv4) have a small gap.

In the actual detection results of TIPYOLO and other methods on the YDTIP dataset, as we can see from the comparison Figure 10, compared with other methods, TIPYOLO can be applied to complex scenes such as those including small objects and occluded objects. In the dense pedestrian scene, other methods have missed detection to different degrees, as shown in the blue dotted box marked in Figure 10. Additionally, EfficientDet-D0 has false positives (i.e., two detection boxes appear for one object), as shown in the black dotted box marked in Figure 10. Since the two-stage method [48] is difficult to deploy on edge computing devices, only the results of one-stage object detection methods [51] are visualized here.

#### 3.3.3. Comparison Experiments with Other Methods on the Public Dataset

To further verify the performance of TIPYOLO, a comparative experiment was conducted with other advanced methods on the UTMD [39] dataset. The comparison results between TIPYOLO and other object detection methods are shown in Table 4.

Compared with Yolox-S, the parameters of TIPYOLO are unchanged; it has a greater advantage in terms of detection accuracy, and the FPS performance is comparable. Compared with the lightweight Yolov4-tiny, although TIPYOLO does not have an advantage in terms of parameters and FPS, it has obvious advantages in terms of detection accuracy. Compared with the lightweight EfficientDet-d0, although the TIPYOLO parameter is nearly 2.3 times larger, it has obvious advantages in terms of accuracy and FPS. TIPYOLO is still comparable to YOLOv3 in MR^−2^ metrics, and slightly higher than YOLOv4 and Faster R-CNN.

In the actual detection results of TIPYOLO and other methods on the UTMD dataset, as we can see from Figure 11, TIPYOLO has better performance for small object detection, while other methods have missed detection to varying degrees.

To further validate the generalization performance of TIPYOLO, we also conducted performance tests on the commonly used KAIST and CVC-14 datasets, which are shown in Table 5. Since the value of FPS changes insignificantly, the statistics of the FPS are not shown in Table 5. As can be seen from the table, our method also achieves decent performance on these two datasets.

To sum up, it can be seen that the improvement in Yolox-S is meaningful, and TIPYOLO balances the detection accuracy, inference speed, and model complexity. Meanwhile, our method has certain generalization performance for different datasets. It is more applicable in thermal imaging pedestrian-detection scenarios.

#### 3.3.4. Evaluation of Method Performance on Edge Computing Devices

To verify the effectiveness of TIPYOLO on edge computing devices, we utilized a dense pedestrian video with 5400 frames for performance testing. The experimental verification platform adopts Cambrian Extrobox202. The machine configuration utilized in this experiment was as follows: the processor was Intel i3-6100U 2-Core; the memory was 4GB; 240GB SSD; the power supply was 12V/4A; and there was active cooling; the NPU adopted an MLU220-M.2 card with support for int16 and int8; the memory was LPDDR4x and 4G 3200MHz; and the power consumption was 8.25w.

After quantification, the detailed test results of the method accuracy are shown in Table 6; the weight parameters are almost unchanged (i.e., only some scale parameters are added), so weight statistics were not performed here. From the table, we can see that the quantization operation can improve the generalization ability of the model to a certain extent, that is, the AP_75_ increases to different degrees. However, the AP_M_ and AP_L_ of “(float32, float16) to (int8, int8)” and “(float32, float32) to (int8, int8)” have decreased. The changes in MR^−2^ are insignificant.

Next, we analyzed the computing speed of the method. TIPYOLO consists of pre-processing, inference, post-processing, and visualization, in which pre-processing, post-processing, and visualization are performed on the CPU and model inference on the NPU. As can be seen from Table 7, the processing speed of Structure a does not meet the real-time requirements (i.e., above 30 FPS), with a maximum of 20.9 FPS. Moreover, we can see from the table the effect of different degrees of quantification on accelerated processing. In the same case, int8 quantization is faster than int16 quantization, which can be seen in Table 7 (rows 1 and 2 or rows 3 and 4). Similarly, float16 is faster than float32 (rows 1 and 3 or rows 2 and 4). The experimental analysis shows that the speed of Structure a is slower than that of Structure b due to the existence of operator serial dependence (columns 2 and 3). In addition, it can be seen from the experimental analysis that the pre-processing operation limits the overall calculation speed of the method, and the multi-threading strategy can effectively alleviate this problem (columns 1, 2, and 3).

For single-channel video processing, our proposed Structure c meets the requirement of scene timeliness. From Figure 12a, we can see that multi-thread stacking increases CPU utilization and also increases the calculation speed of the method. For two-channel video processing, it can be seen from Figure 1a,c that under the int8 + float16 combination strategy, the multi-channel concurrency method is slightly lower than multi-channel parallelism in terms of CPU utilization rate (the multi-channel parallelism CPU is almost fully loaded, and parallelism patterns have one more visualization thread and one more post-processing thread than concurrency), while the MLU core utilization rate is comparable. In general, the parallelism calculation speed is slightly slower, which may be due to an excessive CPU performance load in parallelism mode. Meanwhile, it can be seen from Figure 12b,c that parallelism is faster than concurrency when the MLU core utilization rate is close to the full load state, and it also means high power consumption.

From Figure 12b, we find that a greater degree of quantization has a mitigation effect on the NPU power consumption, and has a speed-up effect. Additionally, from the red and green lines in Figure 12c, we find that a greater degree of quantization can reduce the computational burden on the NPU core.

## 4. Conclusions

In this paper, we propose a lightweight and efficient object detection method to solve the problem of thermal infrared pedestrian detection. Moreover, we verify the effectiveness and robustness of our method through a comparative study. Multi-scale mosaic data augmentation improves the diversity of data and reduces the risk of model overfitting, which simulates body occlusion and multi-scale objects. The parameter-free attention module is introduced between the backbone and neck to enhance feature extraction. To meet the requirements of the single-channel and two-channel video processing business, we combine quantization and multi-threading technology to accelerate method calculation. The experimental results show that our method has certain advantages over other methods in the thermal imaging pedestrian-detection scene, which is effective for false positives and missed detections, and has a high computing speed on edge computing devices.

In future work, we will further focus on model parameter compression and precision tuning. Furthermore, our method will be applied to auxiliary temperature measurement during the COVID-19 epidemic, which achieves automated abnormal body temperature detection and tracking.

## Figures and Tables

**Figure 1 sensors-22-06710-f001:**
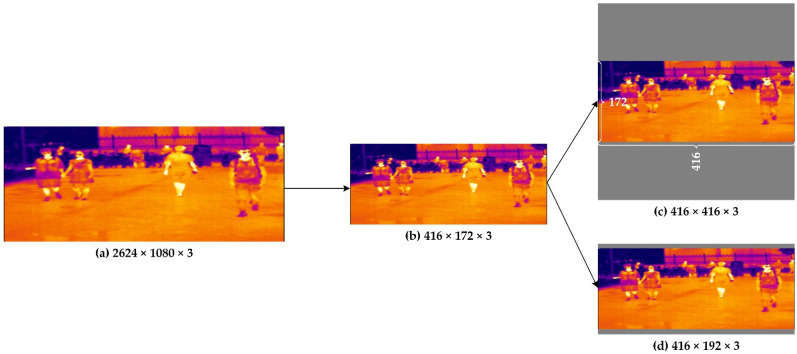
The input image is undistorted. (**a**) the original image; (**b**) the resized image; (**c**) the gray-filled image with a data dimension of 416 × 416 × 3; and (**d**) the gray-filled image with a data dimension of 416 × 192 × 3.

**Figure 2 sensors-22-06710-f002:**
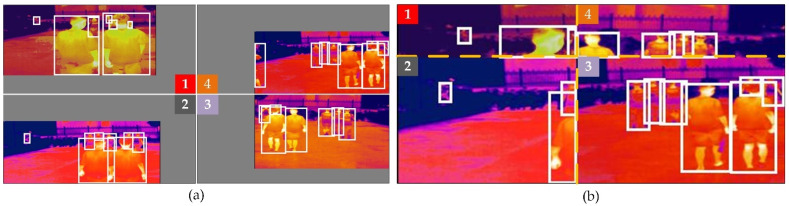
Mosaic data augmentation. (**a**) contains four images, corresponding to 1–4, each of which is processed through the data augmentation strategy described in Section 3.1; (**b**) is an image composed of 1–4 in (**a**), where the white boxes in the figure represent the ground-truth label.

**Figure 3 sensors-22-06710-f003:**
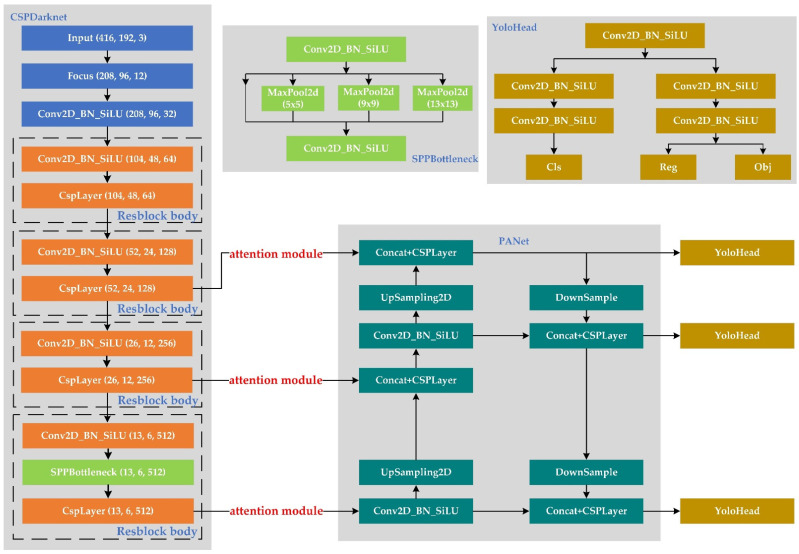
TIPYOLO network structure.

**Figure 4 sensors-22-06710-f004:**
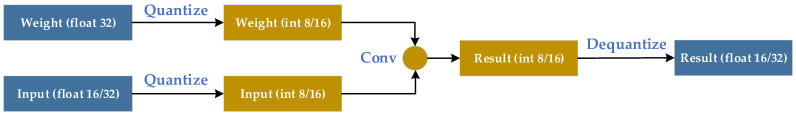
Each network layer’s quantization process.

**Figure 5 sensors-22-06710-f005:**
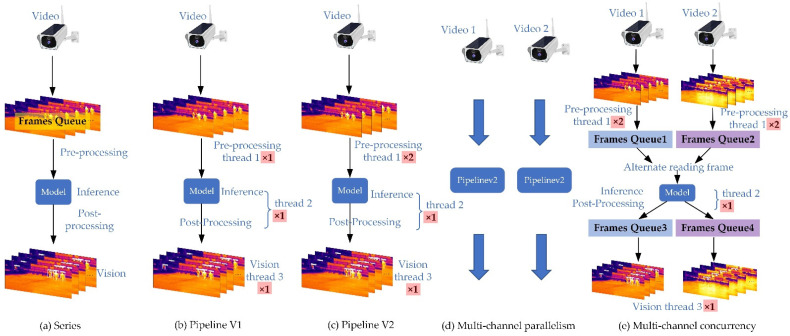
Single-channel and multi-channel video computing acceleration with multi-threading. (**a**) A single thread completes pre-processing, inference, and post-processing; (**b**) pre-processing, inference, and post-processing are completed by a single thread, respectively; (**c**) differs from (**b**) by adding two threads to the pre-processing; (**d**) video parallelism processing using pipelineV2; (**e**) video concurrency processing using the improved pipelineV2.

**Figure 6 sensors-22-06710-f006:**
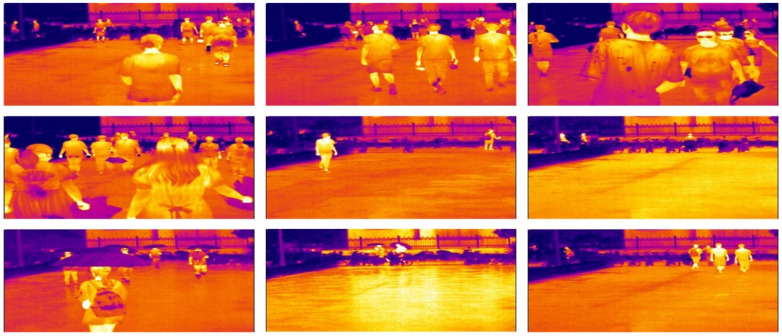
YDTIP dataset visualization results.

**Figure 7 sensors-22-06710-f007:**
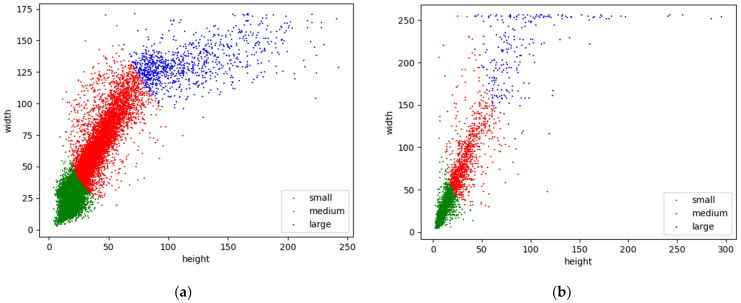
Distribution of large, medium, and small objects in the dataset. (**a**) YDTIP; (**b**) UTMD.

**Figure 8 sensors-22-06710-f008:**
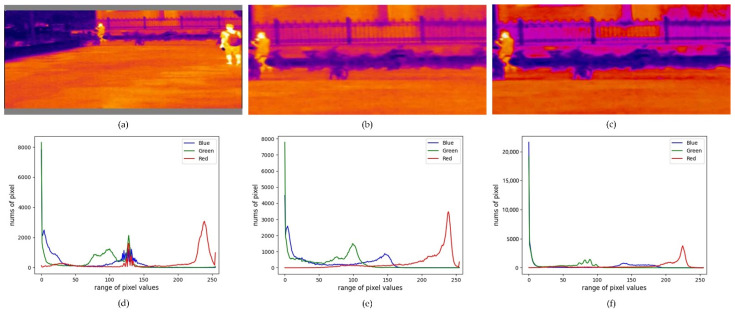
The result of an image being processed through the above data augmentation strategy. (**a**) The original image with gray filling; (**b**) the result of IHF and LWS; (**c**) the result of CGT-HSV, where the data dimensions of the three images are all 416 × 192 × 3; and (**d**–**f**) the pixel histograms of the RGB channels, which correspond to (**a**–**c**).

**Figure 9 sensors-22-06710-f009:**
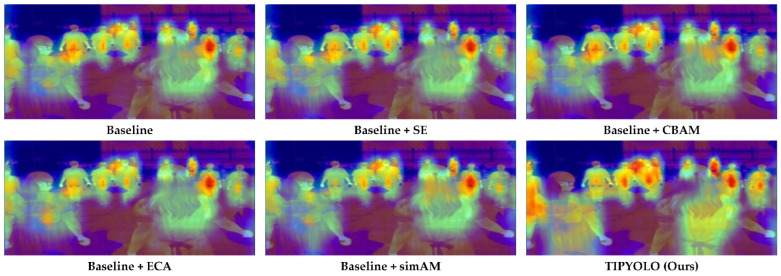
The activation heatmap for each detector generated using GradCAM.

**Figure 10 sensors-22-06710-f010:**
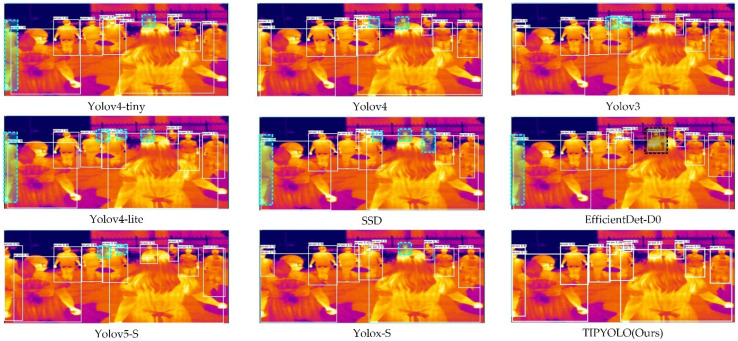
The detection results of YIPYOLO and other methods on the YDTIP dataset. The blue dotted boxes represent missed detections; the black dotted boxes represent false-positive detections.

**Figure 11 sensors-22-06710-f011:**
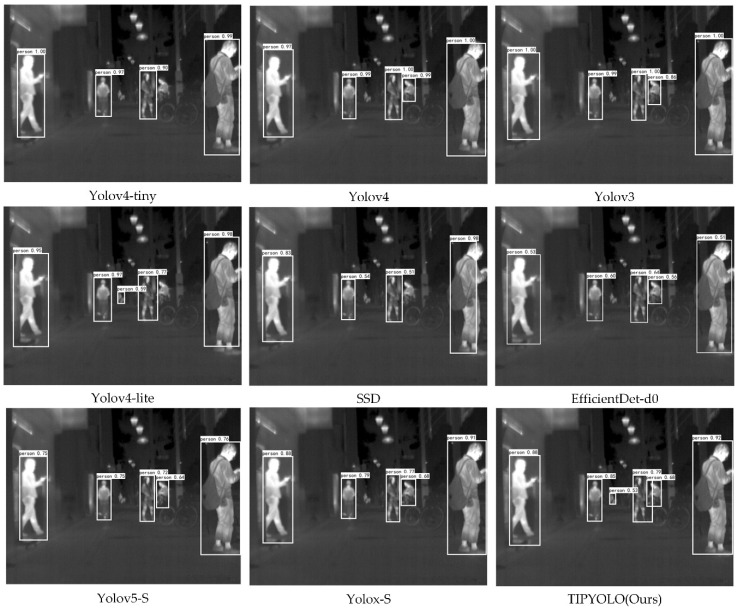
The detection results of YIPYOLO and other methods on the UTMD dataset.

**Figure 12 sensors-22-06710-f012:**
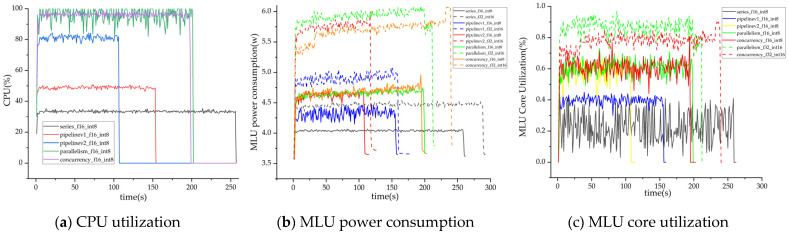
Effect of differently structured methods on hardware utilization and power consumption. (**a**) The change in CPU utilization when the five structures in Figure 5, respectively, are running; (**b**) the effect of different strategies on NPU power consumption; and (**c**) the effect of different strategies on NPU core utilization.

**Table 1 sensors-22-06710-t001:** Experiment results of different input sizes. “↑” indicates that the larger the value, the better the performance.

Size	AP ↑	AP_50_ ↑	AP_75_ ↑	AP_S_ ↑	AP_M_ ↑	AP_L_ ↑
640 × 640	0.701	0.959	0.785	0.413	0.722	0.869
416 × 416	0.654	0.950	0.721	0.494	0.773	0.895
416 × 192	0.657	0.951	0.726	0.501	0.772	0.895

**Table 2 sensors-22-06710-t002:** Results of ablation experiment. “↑” indicates that the larger the value, the better the performance.

Method	Operators	Parameters	AP ↑	AP_50_ ↑	AP_75_ ↑	AP_S_ ↑	AP_M_ ↑	AP_L_ ↑
Baseline	-	-	0.657	0.951	0.726	0.501	0.772	0.895
+ SE	GAP, FC, ReLU	$2(C_{1}^{2}+C_{2}^{2}+C_{3}^{2})/r$	0.659	0.951	0.733	0.505	0.774	0.891
+ CBAM	GAP, GMP, FC, ReLU, CAP, CMP, BN, C2D	$\begin{array}{l} 2(C_{1}^{2}+C_{2}^{2}+C_{3}^{2})/r\\ +2(k_{1}^{2}+k_{2}^{2}+k_{3}^{2}) \end{array}$	0.659	0.950	0.727	0.500	0.776	0.900
+ ECA	GAP, C1D	$k_{1}+k_{2}+k_{3}$	0.661	0.949	0.734	0.504	0.775	0.902
+ simAM	GAP, /, ⊙, +	0	0.659	0.951	0.731	0.502	0.776	0.893
+ Mosaic	-	0	0.663	0.950	0.736	0.509	0.775	0.886
+ Multi-scale mosaic	-	0	0.664	0.951	0.738	0.505	0.779	0.903
+ Multi-scale mosaic + simAM	GAP, /, ⊙, +	0	0.663	0.951	0.736	0.509	0.778	0.897

**Table 3 sensors-22-06710-t003:** The comparison results of the methods on the YDTIP dataset. “↑” indicates that the larger the value, the better the performance; “**↓**” indicates that the smaller the value, the better the performance.

Method	Size	Parameters	AP ↑	AP_50_ ↑	AP_75_ ↑	AP_S_ ↑	AP_M_ ↑	AP_L_ ↑	MR^−2^ ↓	FPS ↑
Faster R-CNN [48]	416 × 416	157.75 M	0.678	0.949	0.760	0.549	0.776	0.851	0.146	22.2
Yolov4-tiny [25]	416 × 416	22.57 M	0.566	0.924	0.614	0.417	0.683	0.724	0.206	160.3
Yolov4 [17]	416 × 416	243.9 M	0.645	0.952	0.724	0.51	0.747	0.806	0.146	41.2
Yolov3 [51]	416 × 416	234.69 M	0.611	0.953	0.68	0.476	0.714	0.801	0.145	54.4
Yolov4-lite [23]	416 × 416	46.79 M	0.576	0.939	0.629	0.434	0.678	0.755	0.193	56.1
SSD [18]	416 × 416	90.58 M	0.573	0.917	0.613	0.405	0.693	0.734	0.275	63.1
EfficientDet-D0 [26]	512 × 512	14.6 M	0.559	0.891	0.614	0.413	0.671	0.808	0.319	25.5
Yolov5-S [27]	416 × 416	26.95 M	0.543	0.9	0.572	0.369	0.676	0.793	0.267	79.7
Yolox-S [30]	416 × 416	34.09 M	0.654	0.95	0.721	0.494	0.773	0.895	0.153	76.6
TIPYOLO (ours)	416 × 192	34.09 M	0.663	0.951	0.736	0.509	0.778	0.897	0.156	75.0

**Table 4 sensors-22-06710-t004:** The comparison results of the methods on the UTMD dataset. “↑” indicates that the larger the value, the better the performance; “↓” indicates that the smaller the value, the better the performance.

Method	Size	Parameters	AP ↑	AP_50_ ↑	AP_75_ ↑	AP_S_ ↑	AP_M_ ↑	AP_L_ ↑	MR^−2^ ↓	FPS ↑
Faster R-CNN [48]	320 × 256	157.75 M	0.539	0.884	0.588	0.452	0.673	0.733	0.27	19.6
Yolov4-tiny [25]	320 × 256	22.57 M	0.352	0.677	0.346	0.222	0.573	0.651	0.447	162.5
Yolov4 [17]	320 × 256	243.9 M	0.439	0.845	0.403	0.343	0.59	0.658	0.286	46.8
Yolov3 [51]	320 × 256	234.69 M	0.42	0.842	0.375	0.322	0.574	0.669	0.296	65.8
Yolov4-lite [23]	320 × 256	46.79 M	0.36	0.757	0.306	0.239	0.533	0.662	0.422	57.5
SSD [18]	320 × 320	90.58 M	0.395	0.757	0.345	0.293	0.547	0.662	0.417	85.5
EfficientDet-D0 [26]	512 × 512	14.6 M	0.366	0.676	0.357	0.278	0.528	0.59	0.537	25.9
Yolov5-S [27]	320 × 256	26.95 M	0.401	0.756	0.38	0.299	0.586	0.67	0.46	78.3
Yolox-S [30]	320 × 256	34.09 M	0.47	0.837	0.472	0.35	0.646	0.788	0.32	75.0
TIPYOLO (ours)	320 × 256	34.09 M	0.484	0.848	0.481	0.369	0.65	0.78	0.298	70.6

**Table 5 sensors-22-06710-t005:** The comparison results of the methods on KAIST and CVC-14 datasets. In columns 2 to 8, the values to the left of “/” represent the test result on the KAIST data, and the values to the right of “/” represent the test result on the CVC-14 dataset. “↑” indicates that the larger the value, the better the performance; “↓” indicates that the smaller the value, the better the performance.

Method	AP ↑	AP_50_ ↑	AP_75_ ↑	AP_S_ ↑	AP_M_ ↑	AP_L_ ↑	MR^−2^ ↓
Faster R-CNN [48]	0.73/0.641	0.954/0.967	0.843/0.756	0.669/0.512	0.698/0.666	0.86/0.744	0.078/0.101
Yolov4-tiny [25]	0.464/0.45	0.92/0.83	0.395/0.448	0.296/0.267	0.439/0.566	0.617/0.633	0.217/0.298
Yolov4 [17]	0.543/0.563	0.963/0.972	0.565/0.605	0.417/0.45	0.536/0.587	0.666/0.655	0.063/0.097
Yolov3 [51]	0.591/0.567	0.971/0.969	0.663/0.619	0.437/0.457	0.58/0.592	0.706/0.662	0.058/0.111
Yolov4-lite [23]	0.529/0.526	0.957/0.954	0.514/0.543	0.385/0.366	0.499/0.563	0.662/0.624	0.103/0.148
SSD [18]	0.631/0.545	0.957/0.939	0.719/0.566	0.463/0.405	0.608/0.582	0.789/0.620	0.101/0.215
EfficientDet-D0 [26]	0.366/0.403	0.799/0.786	0.265/0.362	0.21/0.161	0.347/0.468	0.51/0.605	0.49/0.483
Yolov5-S [27]	0.323/0.51	0.728/0.911	0.21/0.512	0.169/0.322	0.311/0.553	0.455/0.662	0.538/0.283
Yolox-S [30]	0.595/0.552	0.924/0.92	0.655/0.599	0.389/0.391	0.646/0.636	0.806/0.747	0.162/0.206
TIPYOLO(Ours)	0.695/0.614	0.974/0.967	0.808/0.712	0.537/0.461	0.676/0.647	0.825/0.727	0.05/0.112

**Table 6 sensors-22-06710-t006:** Effects of different quantization degrees on the method’s accuracy. In the first column, the first value in the first parentheses represents the numerical type of the weight, and the second value in the first parentheses represents the numerical type of the input. “to” represents converting the numeric type in the first parenthesis to int8 or int16. “↑” indicates that the larger the value, the better the performance; “↓” indicates that the smaller the value, the better the performance.

Strategy	AP ↑	AP_50_ ↑	AP_75_ ↑	AP_S_ ↑	AP_M_ ↑	AP_L_ ↑	MR^−2^ ↓
unquantified	0.663	0.951	0.736	0.509	0.778	0.897	0.156
(float32, float16) to (int16, int16)	0.670	0.951	0.749	0.519	0.782	0.896	0.156
(float32, float16) to (int8, int8)	0.664	0.950	0.745	0.514	0.774	0.888	0.158
(float32, float32) to (int16, int16)	0.670	0.952	0.749	0.519	0.782	0.897	0.156
(float32, float32) to(int8, int8)	0.662	0.952	0.741	0.514	0.771	0.888	0.152

**Table 7 sensors-22-06710-t007:** Effects of different quantization degrees on the method’s FPS. a, b, c, d, and e in the table, respectively, represent the five video processing methods in Figure 5, that is, the strategy corresponding to the five subfigures, where the two methods in strategy d are run in parallel.

Strategy	a	b	c	d		e
(float32, float16) to (int16, int16)	19.6	34.7	48.5	26.3	26.1	49.3
(float32, float16) to (int8, int8)	20.9	35.4	51.2	26.6	26.5	53.4
(float32, float32) to (int16, int16)	18.8	34.0	45.1	26.0	25.2	44.5
(float32, float32) to (int8, int8)	19.2	34.2	47.0	26.2	26.0	46.3

## Data Availability

The data presented in this study are available on request from the corresponding author.

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
