# Peer review of "A Thermal Infrared Pedestrian-Detection Method for Edge Computing Devices"

_sensors, 2022, doi:10.3390/s22176710_

Round 1
Reviewer 1 Report
Main remarks:
1. In my opinion, the methodology of evaluating the results is incorrectly defined. The standard in pedestrian detection is the use of the miss rate with false positive per image and log-average miss-rate metrics, as in:
Ning, C., Menglu, L., Hao, Y. et al. Survey of pedestrian detection with occlusion. Complex Intell. Syst. 7, 577–587 (2021). https://doi.org/10.1007/s40747-020-00206-8
In the pedestrian detection task we have strongly unbalanced sets of positive and negative samples, therefore the miss rate and FPPI metrics are much better suited for evaluating the results.
2. There is no comparison with the results presented in the literature. In addition, the experiments should be supplemented with tests with commonly used databases of thermal imaging recordings, which constitute a comparative standard, i.e. KAIST, CVC-14, as in:
J. Cao, Y. Pang, J. Xie, F. S. Khan and L. Shao, "From Handcrafted to Deep Features for Pedestrian Detection: A Survey," in IEEE Transactions on Pattern Analysis and Machine Intelligence, doi: 10.1109 / TPAMI.2021.3076733.
Minor remarks:
1. Figure 2 is poorly prepared. No explanation as to why it is presented in such colors, no information about the meaning of rectangles. Besides, it should be divided into two lines because it is too wide.
2. Lines 64-64: should be: … huge number of ... in „which have huge parameters”
3. Lines 146-147: The sentence is not clear: „Hence, the batch size data is in- 146 necessary to be too large, and a GPU can achieve better results.”
3. Too often phrase "In this paper" is used.
4. Lines 292-295: These sentences are not clear: „The thermal imaging dataset has a drab object color. If the model is directly trained, it is prone to overfitting. To solve this problem, we added color gamut transformation on the HSV color space, length and width scaling, and image horizontal flipping”
5. This sentence lacks logic, it is not explained why color gamut transformation on the HSV color space is added to infrared images that are monochromatic (they represent the temperature value for a given pixel).
„The thermal imaging dataset has a drab object color. If the model is directly trained, it is prone to overfitting. To solve this problem, we added color gamut transformation on the HSV color space, length and width scaling, and image horizontal flipping.”
6. Line 296: Shouldn't there be YDTIP instead of TIPYOLO?
Summarizing, the text should be described more carefully. Some parts of the explanations are not clear. I would recommend to review of the entire paper carefully.
Author Response
Thanks so much for your comments. Please see the attachment.

Reviewer 2 Report
Well presented manuscript with interesting results and current topic.
I propose to the authors a minor revision. I have missed in the state of the art some initiatory works of Professor Maldague on the subject of using infrared thermal cameras in pedestrian detection.
Author Response

(The authors gave the same response as above.)

Reviewer 3 Report
Please explain the term " edge devices" considering a broad range of authors in one more sentence.
Please make Figure 2 larger. It is difficult to comprehend the image.
Please indicate the level of pedestrian identification. The level of quality as such being seeking is face recognition? Why not include the COVID-19 application in the abstract as well?
How did authors establish the Table 5? Which techniques were used for methods accuracy?
Author Response

(The authors gave the same response as above.)

Round 2
Reviewer 1 Report
Thanks for the detailed answers. I appreciate the work and the introduced changes.
I would suggest adding a comparison with other methods for the presented results for the KAIST and CVC-14 databases.
Author Response

(The authors gave the same response as above.)
